# SMAR: Soft Modality-Aware Routing Strategy for MoE-based Multimodal Large Language Models Preserving Language Capabilities

## Abstract

Mixture-of-Experts (MoE) architectures have become a key approach for scaling large language models, with growing interest in extending them to multimodal tasks. Existing methods to build multimodal MoE models either incur high training costs or suffer from degraded language capabilities when adapting pretrained models. To address this, we propose Soft Modality-Aware Routing (SMAR), a novel regularization technique that uses Kullback–Leibler divergence to control routing probability distributions across modalities, encouraging expert specialization without modifying model architecture or heavily relying on textual data. Experiments on visual instruction tuning show that SMAR preserves language ability at 86.6% retention with only 2.5% pure text, outperforming baselines while maintaining strong multimodal performance. Our approach offers a practical and efficient solution to balance modality differentiation and language capabilities in multimodal MoE models.

## 1 Introduction

The Mixture-of-Experts (MoE) architecture has seen increasingly widespread adoption in large language models (LLMs). Models such as Mixtral 8×7B (Jiang et al., 2024) and Deepseek-V3 (Liu et al., 2024) employ sparse MoE structures to achieve a favorable balance between substantially increased parameter capacity and inference efficiency. This architectural approach has demonstrated superior overall performance and has progressively emerged as the dominant design for LLMs in industrial applications. Therefore, in contemporary multimodal large language models (MLLMs), integrating the MoE architecture which supports substantial parameter scaling while maintaining inference efficiency has become a competitive choice for achieving a higher performance upper bound. Researchers have primarily adopted three approaches to developing MLLMs based on the MoE architecture. The first approach (Li et al., 2024) involves training a MLLM with a MoE architecture from scratch using extensive datasets. However, this training paradigm demands significant computational overhead, constraining both its scalability and broader applicability. The second approach (Lin et al., 2024; Li et al., 2025) involves extending existing dense LLMs into a MoE architecture during multimodal fine-tuning to form multiple experts. However, this strategy frequently results in weak expert specialization due to high parameter redundancy among experts (Huang et al., 2025), while the limited scale and capabilities of the base models further restrict the model performance. The third approach (Fu et al., 2024; Wu et al., 2024) involves extending pre-trained MoE-based LLMs with multimodal capabilities, thus avoiding the computational burden of training from scratch and mitigating constraints from the original model's linguistic capacity. Furthermore, (Lo et al., 2024) indicates that such MoE models, having been pretrained on large-scale text corpora, already exhibit well-differentiated expert knowledge, thereby reducing the likelihood of redundant expert specialization during multimodal adaptation. Therefore, we hold the view that the third approach offers higher feasibility for obtaining MLLMs with MoE architectures.

However, a notable challenge during multimodal transfer training is the potential degradation of the model's language capabilities. Previous works such as VITA (Fu et al., 2024) and DeepSeek-VL2 (Liu et al., 2024) incorporate approximately 20% pure textual data during multimodal training to help preserve the model's language capabilities. Nevertheless, this strategy not only increases the training time cost, but also raises the acquisition cost of high-quality textual data during multimodal

training. Other works (Long et al., 2024) have explored modality expansion by incorporating efficient fine-tuning modules while freezing the backbone of the language model to preserve its original language capabilities. However, due to the limited number of tunable parameters in these modules, the multimodal performance ceiling of the model is often constrained.

Consequently, reducing the reliance on textual data while preserving language capabilities remains a significant challenge in building effective MLLMs. In this paper, we propose a novel modality-aware routing strategy to address this issue. Previous studies (Li et al., 2025) have shown that in MoE-based MLLMs, experts tend to exhibit modality preferences, resulting in notable differences in the probability of routing tokens from different modalities to the same expert. This observation motivates us to explore modality-aware routing strategies that explicitly control modality preferences in the routing mechanism, thereby encouraging the specialization of experts in modality-specific knowledge and ultimately helping to preserve linguistic capabilities. However, most existing MoE-based MLLMs predominantly employ routing under load-balancing loss constraints or resort to manually enforced hard partitioning of modality-specific experts (Luo et al., 2024). This rigid partitioning will split the original knowledge areas of experts, making it difficult to determine the optimal grouping strategy, thus failing to achieve the goal of maintaining strong language performance.

Motivated by the above considerations, we design a statistical method to characterize the routing probability distributions across tokens from different modalities (modality routing distribution, MRD). Based on these distributions, we compute the distance between the routing probability distributions of different modality tokens using the Kullback–Leibler divergence and introduce an auxiliary loss to constrain this divergence. Without any modifications to the data or model architecture, we manually control the modality preferences of the model's experts, which effectively helps to preserve the model's language capabilities. Moreover, unlike conventional finetuning approaches, our method does not require freezing the model backbone to preserve language capabilities, thereby fully unleashing the model's multimodal performance potential.

Our contributions can be summarized as follows:

- We propose a novel metric for evaluating the routing probability distributions of tokens from different modalities, introducing a new perspective for analyzing routing strategies in MoE-based multimodal models.

- Based on the understanding of modality routing probability distributions, we employ the Kullback–Leibler divergence to measure the MRD distance and impose a constraint through the Soft Modality-Aware Routing (SMAR) loss. This method allows explicit control over the degree of expert modality differentiation without requiring any architectural modifications.

- Extensive experiments demonstrate that controlling expert modality differentiation during multimodal training via SMAR reduces the impact of data distribution on expert specialization. SMAR achieves strong multimodal performance and attains a language capability retention rate of 86.6% on visual instruction finetuning data with only 2.5% pure text, outperforming both the baseline without auxiliary loss (81.6%) and the model using load balancing loss alone (82.8%).

## 2 RELATED WORKS

### 2.1 PRESERVING LANGUAGE CAPABILITIES IN MLLMS

Research on maintaining language capabilities in MLLMs is still in its early stages. Most mainstream approaches for preserving language capabilities rely on increasing the proportion of pure-text instruction fine-tuning data, as exemplified by models such as Qwen2-VL (Wang et al., 2024) and DeepSeek-VL2 (Wu et al., 2024). Although freezing the LLM backbone and employing efficient fine-tuning modules such as LoRA (Hu et al., 2022) can endow the model with multimodal capabilities while preserving much of its language proficiency, the limited number of tunable parameters in these methods tends to restrict the model's multimodal performance ceiling, particularly during large-scale training.

In contrast, we seek to explore a more cost-effective approach with a higher multimodal performance ceiling. SMAR leverages the inherent advantages of the MoE architecture by controlling the differen-

tiation of modality preference among experts to store knowledge specific to different modalities. This enables the preservation of language capabilities while maintaining strong multimodal performance.

## 2.2 MoE Routing Strategy in MLLMs

Current research on modality-aware routing strategies is limited. For instance, Mono-InternVL (Luo et al., 2024) uses a rule-based hard routing that maps image and text tokens exclusively to corresponding experts, necessitating extensive visual pretraining data. Similarly, VL-MoE (Shen et al., 2023) separates visual and textual experts in lower layers and fuses them at higher layers, combining modality-specific feature separation with semantic fusion. However, both methods require significant architectural modifications, limiting their applicability to existing MoE-based large language models. Flex-MoE (Yun et al., 2024) proposes a relatively soft modality-distinguished routing strategy by predefining the number of experts corresponding to each modality according to the modality count. It computes a cross-entropy loss based on the tokens' modality labels and their routing probabilities to encourage modality-specific routing. However, this approach still requires manual specification of the number of experts per modality and lacks a global understanding of the overall distribution of routing probabilities as a constraint.

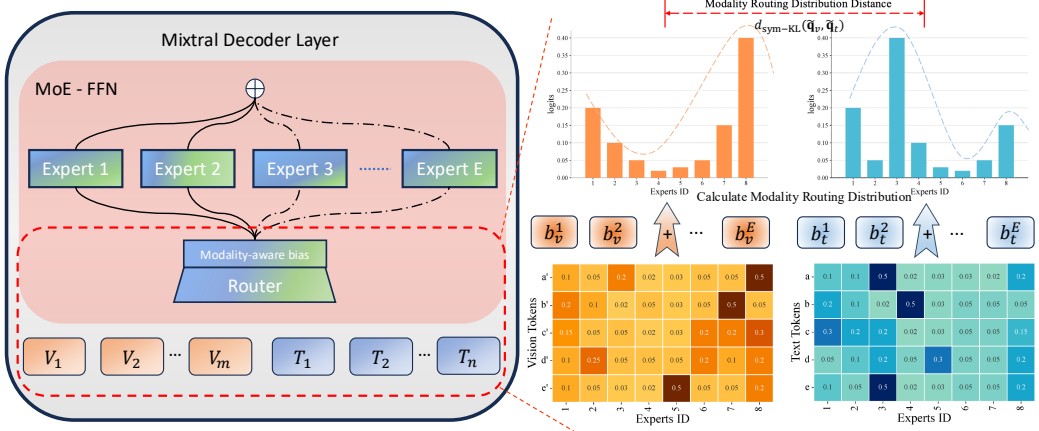

Figure 1: **Illustration of the proposed Soft Modality-Aware Routing (SMAR) mechanism inside a single Mixtral decoder layer. Left:** Visual tokens $\{V_1, \ldots, V_m\}$ (orange) and textual tokens $\{T_1, \ldots, T_n\}$ (blue) share the same router and experts while modality-aware biases are applied to corresponding tokens for soft modality differentiation. $\{b_v^1, \ldots, b_v^E\}$ represents biases for vision tokens and $\{b_t^1, \ldots, b_t^E\}$ for text. The color gradient of experts denote modality preference. **Right:** The token-expert matrix (heat-map) represents the router logits of each token. We calculate the modality routing distribution by our method and the symmetric KL divergence $d_{\text{sym-KL}}(\tilde{\mathbf{q}}_v, \tilde{\mathbf{q}}_t)$ (red bracket) quantifies the cross-modal routing gap and is kept within a tolerance band—by the SMAR loss (Eq. 13).

## 3 METHOD

We propose a soft modality-aware routing strategy for MoE-MLLMs. First, we define the **Modality Routing Distribution(MRD)** to capture routing patterns per modality. Second, we introduce the **Soft Modality-Aware Routing (SMAR)** loss, which uses the KL divergence to regularize the MRD and thereby control experts' modality preferences. Then, we provide an explanation of how this loss is integrated with standard objectives. Finally, we describe the model architecture and the two-stage training strategy.

Our goal is to have some experts specialize in pure language tasks, others focus on vision tasks, and yet others act as multimodal fusion experts handling large amounts of both textual and visual information. We do not explicitly assign which experts should take on each role. Instead, the model autonomously selects and differentiates expert responsibilities under the **soft constraints** imposed by SMAR.

### 3.1 MODALITY–AWARE ROUTING DISTRIBUTION

Consider a mini-batch containing $N$ tokens, among which $N_v$ are visual and $N_t$ are textual ($N = N_v + N_t$). Let $\mathbf{C} \in \mathbb{R}^{N \times H}$ be the hidden states, where $H$ is the hidden dimension. In an MoE Decoder layer such as the one used in Mixtral 8×7B (Jiang et al., 2024), each token is routed to a subset of $E$ feed-forward experts. We denote the router network by $g(\cdot)$ and index experts with $e \in \{1, \ldots, E\}$.

**Router logits.** To explicitly control modality preference, we introduce trainable **modality-aware bias** vectors $\mathbf{b}_v, \mathbf{b}_t \in \mathbb{R}^E$ for vision and text, respectively. The router logits for the two modalities are

$$\mathbf{L}_v = g(\mathbf{C}_v) + \mathbf{1}\, \mathbf{b}_v^\top \in \mathbb{R}^{N_v \times E}, \tag{1}$$

$$\mathbf{L}_t = g(\mathbf{C}_t) + \mathbf{1}\, \mathbf{b}_t^\top \in \mathbb{R}^{N_t \times E}, \tag{2}$$

$$\mathbf{L} = \mathrm{concat}(\mathbf{L}_v, \mathbf{L}_t) \in \mathbb{R}^{N \times E}, \tag{3}$$

where $\mathbf{1}$ is an all-ones column vector whose length matches the number of tokens in the corresponding modality.

**Routing probabilities.** For each token $i$, the softmax over experts yields

$$P_{i,e} = \mathrm{softmax}(\mathbf{L}_{i,:})_e = \frac{\exp(\mathbf{L}_{i,e})}{\sum_{e'=1}^{E} \exp(\mathbf{L}_{i,e'})}. \tag{4}$$

**Top–$K$ selection.** Following sparse MoE practice, we pick the $K$ experts with the largest $P_{i,e}$: $T_i = \{r_1, \ldots, r_K\} \subseteq \{1, \ldots, E\}$. The weights are renormalised within $T_i$,

$$\hat{w}_{i,e} = \begin{cases} \dfrac{P_{i,e}}{\sum_{e' \in T_i} P_{i,e'}}, & e \in T_i, \\ 0, & e \notin T_i. \end{cases} \tag{5}$$

**Frequency and expected weight.** Let $\mathcal{I}_m = \{\, i \mid \text{token } i \text{ is modality } m \,\}$ and $N_m = |\mathcal{I}_m|$. For each modality $m \in \{v, t\}$ we compute

$$F_{m,e} = \frac{1}{K N_m} \sum_{i \in \mathcal{I}_m} \mathbf{1}[e \in T_i], \tag{6}$$

$$R_{m,e} = \frac{1}{N_m} \sum_{i \in \mathcal{I}_m} \hat{w}_{i,e}. \tag{7}$$

**Modality Routing Distribution.** The unnormalised expert mass is $Q_{m,e} = F_{m,e} R_{m,e}$. Normalising over $E$ experts yields the *Modality Routing Distribution (MRD)* .

$$\tilde{Q}_{m,e} = \frac{Q_{m,e}}{\sum_{e'=1}^{E} Q_{m,e'}}, \tag{8}$$

$$\tilde{\mathbf{q}}_m = (\tilde{Q}_{m,1}, \ldots, \tilde{Q}_{m,E}). \tag{9}$$

We write $\tilde{\mathbf{q}}_v$ and $\tilde{\mathbf{q}}_t$ for vision and text, respectively.

### 3.2 SOFT MODALITY-AWARE ROUTING LOSS

The symmetric Kullback–Leibler divergence between the two routing distributions is

$$d_{\text{sym-KL}} = \tfrac{1}{2}\Big( \mathrm{KL}(\tilde{\mathbf{q}}_v \,\|\, \tilde{\mathbf{q}}_t) + \mathrm{KL}(\tilde{\mathbf{q}}_t \,\|\, \tilde{\mathbf{q}}_v) \Big), \tag{10}$$

$$\mathrm{KL}(\tilde{\mathbf{q}}_v \,\|\, \tilde{\mathbf{q}}_t) = \sum_{e=1}^{E} \tilde{Q}_{v,e} \log \frac{\tilde{Q}_{v,e}}{\tilde{Q}_{t,e}}, \tag{11}$$

$$\mathrm{KL}(\tilde{\mathbf{q}}_t \,\|\, \tilde{\mathbf{q}}_v) = \sum_{e=1}^{E} \tilde{Q}_{t,e} \log \frac{\tilde{Q}_{t,e}}{\tilde{Q}_{v,e}}. \tag{12}$$

We impose a tolerance band $[d_{\min}, d_{\max}]$ on $d_{\text{sym-KL}}$ and penalise violations via the *Soft Modality-Aware Routing (SMAR)* loss:

$$\mathcal{L}_{\text{SMAR}} = \begin{cases} d_{\min} - d_{\text{sym-KL}}, & d_{\text{sym-KL}} < d_{\min}, \\ d_{\text{sym-KL}} - d_{\max}, & d_{\text{sym-KL}} > d_{\max}, \\ 0, & \text{otherwise.} \end{cases} \tag{13}$$

### 3.3 OVERALL TRAINING OBJECTIVE

The final loss combines the primary task loss $\mathcal{L}_{\text{main}}$, the standard load-balancing loss $\mathcal{L}_{\text{balance}}$ (Fedus et al., 2022), and the proposed SMAR loss:

$$\mathcal{L}_{\text{total}} = \mathcal{L}_{\text{main}} + \alpha\,\mathcal{L}_{\text{balance}} + \beta\,\mathcal{L}_{\text{SMAR}}, \tag{14}$$

where $\alpha$ and $\beta$ are hyper-parameters controlling the relative strength of the auxiliary terms.

### 3.4 MODEL ARCHITECTURE

We inherit the overall design of VITA (Fu et al., 2024) but restrict the modalities to vision and text owing to computational constraints. The language backbone is Mixtral 8×7B (Jiang et al., 2024) while the vision branch is instantiated with InternViT-300M (Chen et al., 2024) at an input resolution of 448 px. For high–resolution images, we adopt VITA's dynamic tiling strategy, partitioning each image into non–overlapping 448 px tiles. Every tile is encoded into a sequence of visual tokens, which are subsequently linearly projected by a two–layer MLP connector and concatenated with the textual tokens before being fed into the language model.

### 3.5 TRAINING STRATEGY

Following the two-stage curriculum popularized by LLaVA-1.5 (Liu et al., 2023b), we first perform *visual alignment*, where the language backbone and visual encoder are frozen and only the MLP connector is trained to align visual and textual token representations. Next, during *visual instruction tuning*, the visual encoder remains fixed while both the language backbone and connector are fine-tuned on multimodal instruction data to improve instruction-following ability. The composition of the training corpus and additional implementation details are provided in Section 4.

Table 1: **Comparison among different LVLMs on multimodal benchmarks and language benchmarks.** "Res.","Act.","V","P","M" respectively represent the input image resolution,activated parameters,Vicuna (Chiang et al., 2023),Phi-2 (Javaheripi et al., 2023),Mixtral (Jiang et al., 2024). The best results and second best results are indicated by **boldface** and underline, respectively.

| Method | LLM | Act. | Res. | Multimodal Capabilities | | | | | | | | | Language Capabilities | | | | | | | |
|---|---|---|---|---|---|---|---|---|---|---|---|---|---|---|---|---|---|---|---|---|
| | | | | VQA$^{v2}$ | GQA | VizWiz | SQA$^{I}$ | VQA$^{T}$ | POPE | MME | MMB | MM-Vet | MMLU | C-EVAL | GSM8K | BBH | ARC_c | MBPP | HumanEval | IFEval |
| *Base Model* | | | | | | | | | | | | | | | | | | | | |
| Vicuna-7B (Chiang et al., 2023) | V-7B | 7B | - | - | - | - | - | - | - | - | - | - | 47.4 | 36.7 | 23.4 | 41.4 | 39.3 | 13.8 | 19.5 | 40.8 |
| Vicuna-13B (Chiang et al., 2023) | V-13B | 13B | - | - | - | - | - | - | - | - | - | - | 53.9 | 35.0 | 38.1 | 50.1 | 52.5 | 3.6 | 16.5 | 50.3 |
| Phi-2 (Javaheripi et al., 2023) | P-2.7B | 2.7B | - | - | - | - | - | - | - | - | - | - | 58.5 | 30.6 | 61.6 | 59.3 | 53.2 | 49.2 | 30.5 | 27.7 |
| Mixtral 8×7B (Jiang et al., 2024) | M 8×7B | 13B | - | - | - | - | - | - | - | - | - | - | 72.0 | 55.0 | 67.2 | 68.8 | 82.0 | 49.0 | 23.2 | 22.2 |
| *Dense Model* | | | | | | | | | | | | | | | | | | | | |
| LLaVA-1.5 (Liu et al., 2023b) | V-7B | 7B | 336 | 78.5 | 62.0 | 50.0 | 66.8 | 58.2 | 85.9 | 1510.7 | 64.3 | 30.5 | 46.3 | 22.7 | 19.5 | 41.5 | 30.9 | - | 17.7 | 39.4 |
| LLaVA-1.5 (Liu et al., 2023b) | V-13B | 13B | 336 | 80.0 | 63.3 | 53.6 | 71.6 | 61.3 | 85.9 | 1531.3 | 67.7 | 35.4 | 51.7 | 19.1 | 34.2 | 48.0 | 38.3 | - | 21.3 | 48.9 |
| *Sparse Model* | | | | | | | | | | | | | | | | | | | | |
| MoE-LLaVA-2.7B×4-Top2 (Lin et al., 2024) | P-2.7B | 3.6B | 384 | 79.9 | 62.6 | 43.7 | 70.3 | 57.0 | 85.7 | 1431.3 | 68.0 | 35.9 | 49.0 | 30.2 | 51.7 | 52.5 | 70.9 | 43.4 | 51.2 | 35.4 |
| Baseline | M 8×7B | 13B | 448 | **82.5** | 62.2 | 53.7 | 74.6 | 69.6 | **86.8** | 1634.7 | 72.0 | 32.9 | 67.6 | **47.4** | **57.1** | **62.0** | 77.0 | 10.4 | 46.3 | 48.7 |
| Baseline w/ $\mathcal{L}_{\text{balance}}$ | M 8×7B | 13B | 448 | **82.5** | **62.5** | 55.0 | 74.5 | **69.8** | 86.4 | 1600.6 | 72.4 | **39.4** | 67.8 | 45.9 | 56.6 | 60.2 | **81.0** | 14.8 | 43.9 | 47.5 |
| Baseline w/ SMAR | M 8×7B | 13B | 448 | 82.4 | 62.4 | **55.1** | **75.5** | 69.2 | 86.6 | **1638.8** | **72.7** | 35.9 | **68.0** | 46.8 | 57.0 | 61.8 | 79.3 | **28.4** | **49.4** | **50.7** |

## 4 EXPERIMENTS

### 4.1 EXPERIMENTAL SETUP

**Datasets.** Following MoE-LLaVA (Lin et al., 2024), we use the pretrained data of LLaVA 1.5-558k (Liu et al., 2023b) for the visual alignment stage. And we use the datasets from MIMIC-IT (Li et al., 2023a), LRV (Liu et al., 2023a), SViT (Zhao et al., 2023), LVIS (Wang et al., 2023) and LLaVA-mix-665k (Liu et al., 2023b) for the instruction tuning stage. The proportion of text-only data in visual instruction tuning stage is only 2.5%. More information is detailed in the appendix A.

**Training implementation.** We adopt a two-stage training protocol. In Stage 1, the model is trained with a batch size of 128, a learning rate of 5e-4. Stage 2 uses a larger batch size of 256 and a reduced learning rate of 2e-5 with the same scheduling strategy. The SMAR loss parameters $[d_{\min}, d_{\max}]$ are applied starting from Stage 2, set to [1.5, 2.0], with $\beta = 0.01$. Additional training configurations and hyperparameters are detailed in the appendix A.

### 4.2 EVALUATION DETAILS

We evaluate the multimodal capabilities of our model across a diverse set of multimodal tasks. For general multimodal question answering (QA), we benchmark performance on VQA-v2 (Goyal et al., 2017), MME (Fu et al., 2023), and ScienceQA-IMG (Lu et al., 2022). To evaluate the optical character recognition (OCR) capabilities, we use TextVQA (Singh et al., 2019) and VizWiz (Gurari et al., 2018). For reasoning and fine-grained visual understanding, we evaluate on GQA (Hudson & Manning, 2019), MM-Vet (Yu et al., 2023), and MMBench (Liu et al., 2023c). Additionally, we employ the POPE (Li et al., 2023b) benchmark to measure the model's propensity for hallucination.

We also use a diverse set of benchmarks to evaluate the language capabilities of the proposed model. These include evaluations of general knowledge (MMLU (Hendrycks et al., 2020), C-EVAL (Huang et al., 2023)), mathematical (GSM8K (Cobbe et al., 2021)) and reasoning abilities (BBH (Suzgun et al., 2022), ARC-Challenge (Clark et al., 2018)). Moreover, we evaluate the coding proficiency (MBPP (Austin et al., 2021), HumanEval (Chen et al., 2021)) and instruction-following capabilities (IFEval (Zhou et al., 2023)). All language capability evaluations are performed using the OpenCompass toolkit.

### 4.3 RESULTS

**Multimodal Performance.** As shown in Table 1, our model demonstrates strong multimodal capabilities, largely outperforming LLaVA-1.5-13B (Liu et al., 2023b), a model with a comparable number of activated parameters, across a comprehensive suite of benchmarks. Specifically, on SQA$^I$, MME, MMBench, MM-Vet, VQA$^T$, and VQA$^{v2}$, our model achieves performance gains of 5.4%, 7.0%, 7.4%, 12.8%, and 3.0%, respectively, over LLaVA-1.5-13B. This robust performance underscores its proficiency in handling common multimodal tasks, including general visual question answering, optical character recognition, and understanding scene relationships. Furthermore, when compared against other approaches employing identical datasets and model architectures, SMAR also gets the best results on several metrics across VizWiz, SQA$^I$, MME, and MMBench.

**Preservation of Language Capabilities.** As shown in Table 1, our SMAR method achieves leading performance on benchmarks such as MMLU, MBPP, HumanEval, and IFEval, and attains competitive (second-best) performance on other evaluated tasks.

To isolate the models' language capabilities from gains stemming from instruction tuning, we average performance exclusively across six benchmarks (C-EVAL, MMLU, GSM8K, ARC-Challenge, BBH, and MBPP) that have minimal impact on instruction-following capability to compute the retention ratio of language capabilities, as shown in Table 2. With 6.0% of its instruction-tuning corpus consisting of pure-text prompts, LLaVA-1.5-13B retains 82.0% of the backbone's original language capabilities.

Using only 2.5% pure-text data, SMAR still preserves 86.6%—clearly surpassing both the no-auxiliary-loss variant (81.6%) and the load-balancing-only variant (82.8%).

Table 2: **Language capability retention ratio comparison among different methods**.

| Method | LLM | MMLU | CEVAL | GSM8K | BBH | ARC_c | MBPP | Avg. |
|--------|-----|------|-------|-------|-----|-------|------|------|
| Vicuna-13B | V-13B | 53.9 | 35.0 | 38.1 | 50.1 | 52.5 | 3.6 | 38.9 |
| LLaVA-1.5 | V-13B | 51.7 | 19.1 | 34.2 | 48.0 | 38.3 | 0.0 | 31.9 |
| *Retention ratio %* | | 95.9 | 54.6 | 89.8 | 95.8 | 73.0 | 0.0 | 82.0 |
| Mixtral 8x7B | M 8x7B | 72.0 | 55.0 | 67.2 | 68.8 | 82.0 | 49.0 | 65.7 |
| Baseline | M 8x7B | 67.6 | 47.4 | 57.1 | 62.0 | 77.0 | 10.4 | 53.6 |
| *Retention ratio %* | | 93.9 | **86.2** | **85.0** | **90.1** | 93.9 | 21.2 | 81.6 |
| Baseline w/ $\mathcal{L}_{\text{balance}}$ | M 8x7B | 67.8 | 45.9 | 56.6 | 60.2 | 81.0 | 14.8 | 54.4 |
| *Retention ratio %* | | 94.2 | 83.5 | 84.2 | 87.5 | **98.8** | 30.2 | 82.8 |
| Baseline w/ SMAR | M 8x7B | 68.0 | 46.8 | 57.0 | 61.8 | 79.3 | 28.4 | 56.9 |
| *Retention ratio %* | | **94.4** | 85.1 | 84.8 | 89.8 | 96.7 | **58.0** | **86.6** |

Table 3: **Comparison among different methods applied on MoE-LLaVA.** [†] represent that we reproduced the training of MoE-LLaVA following original settings. "w/ SMAR" means that we apply SMAR loss to MoE-LLaVA.

| Method | Multimodal Capabilities | | | | | | | | | Language Capabilities | | | | | | |
|--------|------|------|--------|------|------|------|------|-----|--------|------|-------|-------|------|-------|------|-----------|
| | VQA$^{v2}$ | GQA | VizWiz | SQA$^I$ | VQA$^T$ | POPE | MME | MMB | MM-Vet | MMLU | C-EVAL | GSM8K | BBH | ARC_c | MBPP | HumanEval |
| Phi-2 (Javaheripi et al., 2023) | - | - | - | - | - | - | - | - | - | 58.5 | 30.6 | 61.6 | 59.3 | 53.2 | 49.2 | 30.5 |
| MoE-LLaVA-2.7B×4-Top2 (Lin et al., 2024) | **79.9** | **62.6** | **43.7** | **70.3** | **57.0** | **85.7** | **1431.3** | 68.0 | **35.9** | 49.0 | 30.2 | 51.7 | 52.5 | 70.9 | **43.4** | 51.2 |
| MoE-LLaVA-2.7B×4-Top2$^†$ | 78.9 | 61.9 | 38.0 | **70.3** | 55.2 | **85.7** | 1402.1 | **68.3** | 34.2 | 52.5 | 30.3 | 53.5 | 52.4 | 72.9 | 40.6 | **52.4** |
| w/ SMAR | 78.9 | 60.7 | 40.3 | **70.3** | 56.3 | 84.5 | 1420.0 | 67.6 | 35.4 | **53.7** | **31.9** | **55.4** | **53.1** | **73.2** | 41.0 | 51.8 |

Notably, in code-related evaluations, SMAR demonstrates substantial improvements: its MBPP performance is nearly double that of the model trained with only load-balancing loss. Concurrently, SMAR also outperforms configurations with only load-balancing loss and with no auxiliary losses by 12.5% and 6.7% on HumanEval as shown in Table 1, respectively. In particular, our multimodal models' backbone is initialized from a base model *without* prior instruction tuning and the preservation of code formatting is likely correlated with instruction-following capabilities. We hypothesize that SMAR enhances instruction-following capability. This is supported by a 6.7% improvement on the IFEval benchmark for SMAR compared to using only load-balancing loss. These improvements may stem from the relatively stringent lower bound we impose on the modal routing distribution distance within SMAR, which encourages modality-specific expert specialization and enhances their sensitivity to linguistic cues in instructions.

As shown in Table 1, LLaVA-1.5 struggles to adhere to specified code formats from examples, resulting in a failure to score on the MBPP benchmark. On knowledge-intensive benchmarks like C-EVAL, its performance drops substantially; LLaVA-1.5-13B, for example, retains only 54.5% of its base model's performance, a decline potentially attributable to the limited proportion of Chinese data. For complex reasoning tasks, such as ARC-Challenge (ARC_c), it preserves merely 73.0% of its original performance. Dense models such as LLaVA-1.5, where text-only instruction fine-tuning data constitutes a small fraction (e.g., 6.0%) of the training corpus, often exhibit significant degradation in language capabilities.

In contrast, our approach utilizes the MoE architecture. When employing only the standard load-balancing loss, performance on ARC-Challenge remains nearly on par with the original base model. When trained without specific auxiliary losses aimed at language preservation, this MoE architecture inherently demonstrates greater resilience to language capabilities degradation from multimodal inputs. However, coding ability is notably harmed under this basic setup, with MBPP performance dropping to 30.2%. The introduction of our SMAR method yields a near two-fold improvement on MBPP, effectively alleviating the issue of code format adherence.

**Generalisability of SMAR.** To validate the generalizability of SMAR across different architectures, we integrate it into MoE-LLaVA. Results are reported in Table 3. To minimize interference from extraneous factors, we build upon the publicly released weights from MoE-LLaVA's first-stage connector and their second-stage visually instruction-finetuned model. Our training is conducted

exclusively in the third stage as defined in their work, focusing solely on the MoE expansion process by training only the model's FFN experts and gating network.

We trained two versions of the model: one strictly following the original MoE-LLaVA training protocol, and the other incorporating the SMAR loss during the third training stage. When applying SMAR to MoE-LLaVA, we set $d_{min}$ to 1.0 and $d_{max}$ to 1.5 encourage modality-based expert differentiation, with the weighting factor $\beta$ set to 0.01 and all other components remained unchanged.

MoE-LLaVA, in its multimodal training, employs an upgrade strategy where FFN layers from the original dense base model are fully replicated for its experts. Theoretically, each such expert FFN retains the full knowledge of the precursor language model, leading to comparatively minor degradation in language cabilities.

Experimental results demonstrate that SMAR effectively preserves language capabilities, achieving the best performance across multiple benchmarks including MMLU, C-EVAL, GSM8K, BBH, and ARC-Challenge. Although its multimodal performance metrics do not fully match those reported for MoE-LLaVA, comparison with our reproduced MoE-LLaVA experiments indicates that SMAR contributes to improvements in certain aspects of multimodal performance.

## 4.4 ABLATION STUDY

Table 4: **Ablation of different lower bound($d_{min}$) and upper bound($d_{max}$) settings.**

| $d_{min}, d_{max}$ | Multimodal Capablities | | | | | Language Capabilities | | | | |
|---|---|---|---|---|---|---|---|---|---|---|
| | MME | SQA | TextQA | GQA | MMB | MMLU | GSM8K | BBH | MBPP | HumanEval |
| 0.1, 0.5 | 1606.1 | 73.7 | **70.0** | 62.1 | 71.2 | 65.6 | 56.8 | **62.3** | 15.8 | 45.7 |
| 0.5, 1.0 | 1622.6 | 73.3 | 69.3 | 62.4 | 72.5 | 66.3 | **58.2** | 62.2 | 9.2 | 47.6 |
| 1.0, 1.5 | 1636.7 | 74.4 | 69.7 | **62.5** | **72.9** | 67.0 | 54.1 | 61.5 | **36.4** | 46.3 |
| 1.5, 2.0 | **1638.8** | **75.5** | 69.2 | 62.4 | 72.7 | **68.0** | 57.0 | 61.8 | 28.4 | **49.4** |

**Ablation on SMAR Thresholds.** We investigate the influence of the $d_{min}$ and $d_{max}$ by evaluating several pairs of values. The results are summarised in Table 4. The best overall language score is obtained for $d_{min} = 1.5$ and $d_{max} = 2.0$.

To gain intuition, we visualise the layer–wise MRD distance for each threshold setting in Figure 2a. The MRD distance is computed from the 2,300 evaluation samples in the MME benchmark. The most notable change occurs in the maximum MRD distance, which increases significantly as the threshold range expands. However, when the lower bound of the SMAR threshold is set too high, the distribution curve of the MRD distance shows little variation, and the difference in mean values diminishes. This may be due to the excessively stringent requirement for expert modality differentiation, which is difficult to achieve through training alone.

We further compare the MRD of the best SMAR model with two baseline variants that do not employ SMAR as shown in Figure 2b. Clear changes in routing strategy emerge. In Figure 3 we plot the proportion of image and text tokens routed to each expert at every layer. After activating SMAR, several experts develop pronounced modality preferences. For instance, Expert 8 in Layer 13 almost exclusively processes text tokens, as shown in Figure 3b.

We also find that the routing collapse occurs on the model that is applied with the lowest thresholds ($d_{min} = 0.1, d_{max} = 0.5$). As shown in Figure 3c, the tokens tend to be routed to the same expert, leading to the worst performance.

**Effect of the Trainable Modality-Aware Bias and the Load–Balancing Loss.** Table 5 presents an ablation in which the trainable modality-aware bias and the conventional load-balancing loss are toggled on and off while keeping the SMAR thresholds fixed. Adding modality-aware bias consistently improves performance. Conversely, introducing the load–balancing loss degrades the results, which explains why we omit it in the final model.

The corresponding MRD distance plots are provided in Figure 2c. A modality-aware bias slightly lowers token-modality separation, whereas the load-balancing loss results in a decrease in the minimum MRD distance of the model.

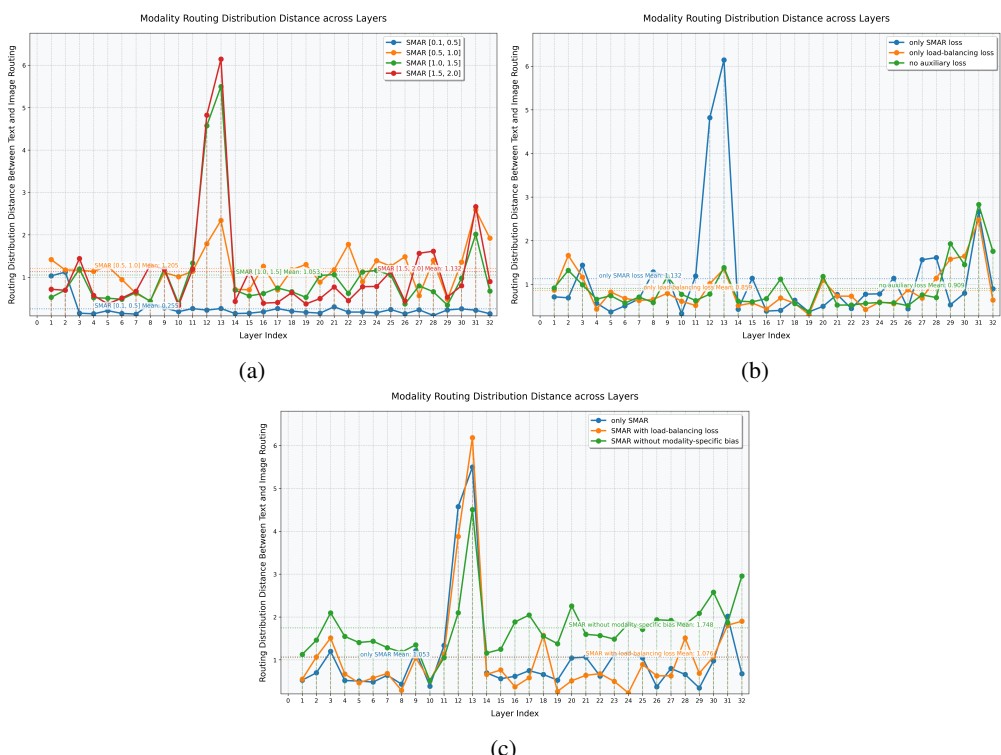

Figure 2: (a) The MRD distance curve of different $[d_{min}, d_{max}]$ settings. We observe that the MRD curves exhibit significant changes in response to different threshold settings. (b) The MRD distance curve of different methods. It is evident that after applying the SMAR method to encourage modality-specific expert differentiation, the MRD curves differ significantly from those observed in methods without SMAR. (c) The MRD distance curve illustrating the effects of applying Modality-Specific Bias and the load-balancing loss within the SMAR framework.

Table 5: **Abalation of modality-specific bias and load-balancing loss on SMAR.**

| $d_{min}, d_{max}$ | Modality–Aware Bias | Load-Balancing Loss | MMLU | GSM8K | BBH | ARC_c | MBPP |
|---|---|---|---|---|---|---|---|
| 1.0, 1.5 | No | No | 63.9 | 54.5 | 62.0 | 79.7 | 17.4 |
| 1.0, 1.5 | Yes | No | **67.0** | 54.1 | 61.5 | 80.3 | **36.4** |
| 1.0, 1.5 | Yes | Yes | 65.0 | **56.0** | 61.7 | **81.0** | 18.8 |

In the appendix, we provide a detailed description of expert activation paths under different algorithms. Based on the differences in expert activation paths, we analyze the potential modality balancing advantages of SMAR. Additionally, ablation studies on various MRD control methods are included, with results supporting our claim that appropriately enhancing modality differences in routing is beneficial for achieving optimal multimodal performance. Furthermore, a range of additional analyses can be found in the appendix.

## 5 CONCLUSION

In this work, we propose a novel perspective, **MRD** for analyzing the routing behavior of different modality tokens in MoE-MLLMs. Building upon this, we introduce the **SMAR** to regulate the degree of modality differentiation among experts. By encouraging modality-specific expert specialization, our method acquires strong multimodal performance and achieves improved preservation of language capabilities without additional pure text data or freezing the backbone.

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

## A  DATASETS AND TRAINING DETAILS

| Hyper-parameter | Stage 1 | Stage 2 |
|---|---|---|
| batch size | 128 | 256 |
| learning rate | 5e-4 | 2e-5 |
| learning rate schedule | cosine | cosine |
| learning rate warm-up ratio | 0.03 | 0.03 |
| weight decay | 0 | 0 |
| grad norm clipping | 1.0 | 1.0 |
| epoch | 1 | 1 |
| optimizer | AdamW | AdamW |
| float precision | bfloat16 | bfloat16 |
| $d_{min}, d_{max}$ | None | 1.5, 2.0 |
| $\alpha$ | None | 0 |
| $\beta$ | None | 0.01 |
| deepspeed configuration | zero3 | zero3 |

Table 6: Hyper-parameter for training.

As shown in Table 7, we merge the datasets used in stages 2 and 3 of MoE-LLaVA into a single training stage. The proportion of pure-text data in stage2 is only 2.5%. All models were trained on Nvidia A800 GPUs.

| Phase | Source | #Sample |
|-------|--------|---------|
| Stage I | LLaVA-1.5-558k | 558k |
| Stage II | SViT-157k,LVIS-220k,LRV-331k,MIMIC-IT-256k, LLaVA 1.5-mix-665k | 1.6M |

Table 7: Composition of the training datasets.

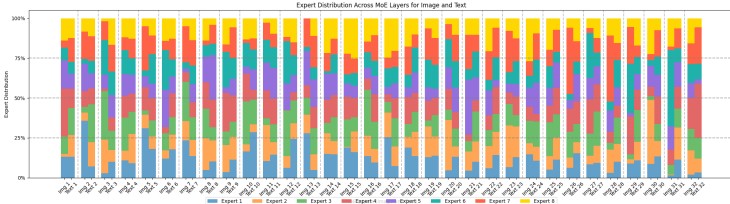

(a) The experts exhibit naturally emerging modality preferences with load-balancing loss.

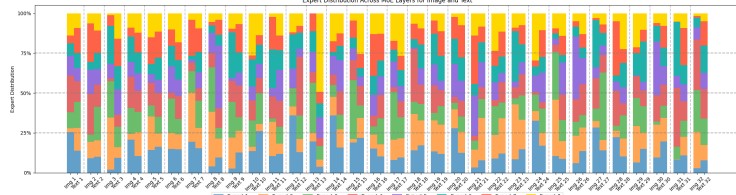

(b) With the $[d_{min}, d_{max}]$ set to [1.5, 2.0], many experts across multiple layers exhibit more pronounced modality preferences—for instance, Expert 8 in layer 13 serves almost exclusively text tokens.

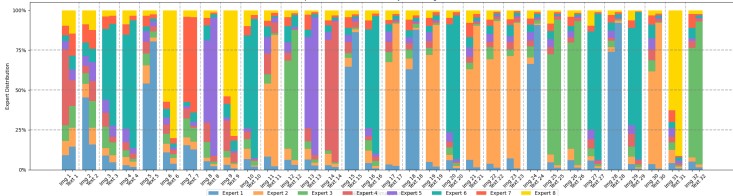

(c) When the threshold is set to [0.1, 0.5], severe routing collapse occurs in all layers starting from layer 3.

Figure 3: **The detailed depiction of the proportion of image and text tokens routed to each expert at every layer.** (a) illustrates the modality preferences of experts across layers in the model trained solely with the load-balancing loss. (b) demonstrates the effectiveness of the SMAR method in controlling expert modality preferences. (c) shows that setting the threshold too low can lead to routing collapse in the model.

# B   MORE ANALYSIS ABOUT ROUTING STRATEGY

**Experts Activated Pathways.**   As shown in Figure 4,Figure 5,Figure 6, we observe that the model trained with the SMAR loss exhibits more pronounced modality preferences in its activated paths. The experts activated pathways is computed from the evaluation samples in the MME benchmark.

Additionally, the coherence of activated paths across modalities is reduced, which may facilitate the model in preserving its original language capabilities.

**Experts Usage Proportion.**   To present more detailed experimental results, we provide visualization outcomes of the utilization rates for all experts, as well as fine-grained utilization statistics for each expert distinguished by modality at every layer. It can be observed that the SMAR loss influences

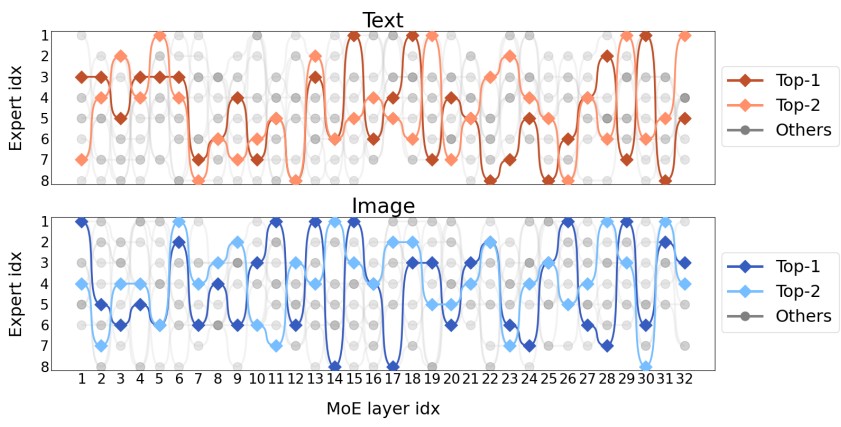

Figure 4: The experts activated pathways without any auxiliary loss.

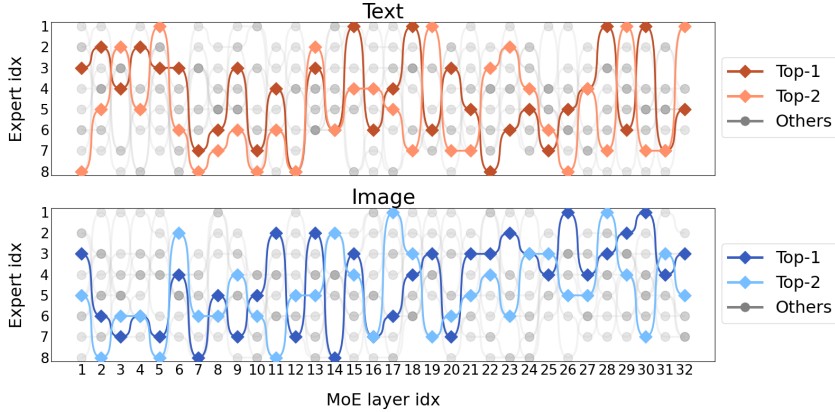

Figure 5: The experts activated pathways with load-balancing loss.

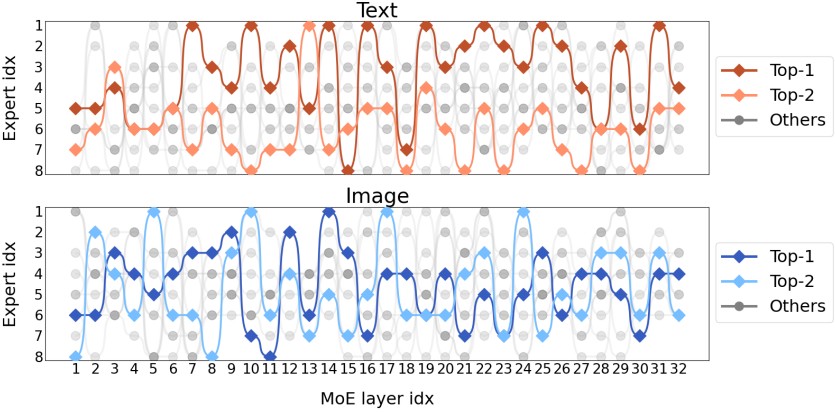

Figure 6: The experts activated pathways with the SMAR bound $[d_{min}, d_{max}]$ set to [1.5, 2.0].

the original modality-specific differentiation process of the experts, thereby altering the evolutionary progression of expert modality specialization, as shown in Figure 18. The experts' usage proportion is computed from the evaluation samples in the MME benchmark.

**Experts Activated on Different Tasks.** To investigate the expert activation patterns of the model trained with the SMAR loss across different tasks, we sampled 200 inference results from each

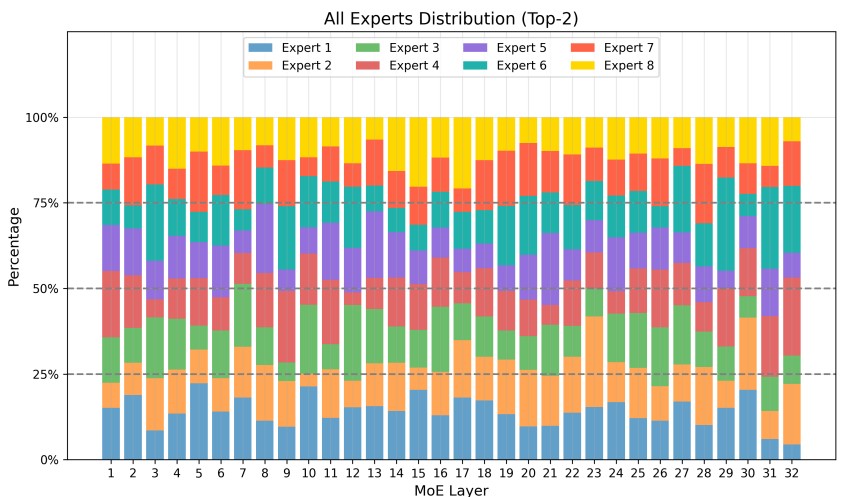

Figure 7: All Experts Usage Proportions for Baseline VITA model

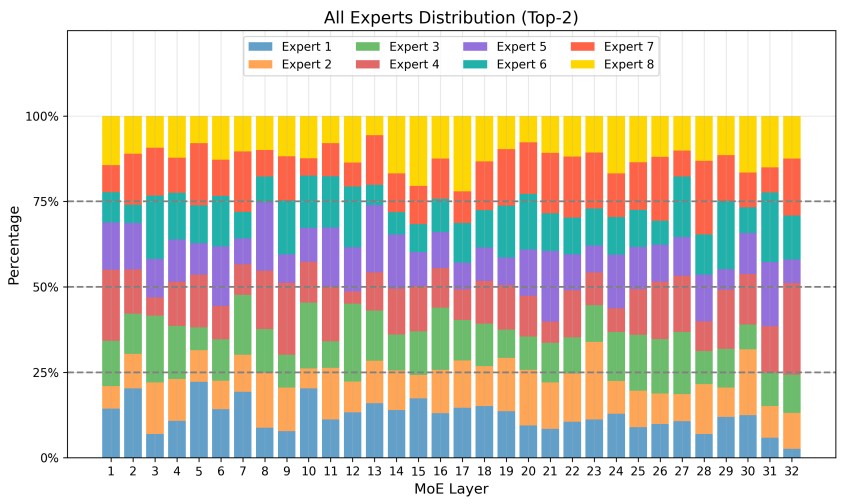

Figure 8: All Experts Usage Proportions for Baseline VITA model with load-balancing loss.

benchmark task other than MME, including SQA-IMG, POPE, and TextVQA. This sampling enabled us to analyze the distribution of expert activations and activated paths across different modalities. As illustrated in Figure 10, Figure 11, Figure 12, the activation levels of experts across different modalities are similar, indicating that the experts exhibit stable modality preference characteristics.

As shown in Figure 13, Figure 14, Figure 15, significant differences are observed in the expert activation paths across different tasks, indicating that the model is capable of selecting experts with task-appropriate knowledge. This elucidates why the model achieves an optimal performance balance across various modalities and tasks.

## C  DISCUSSION ON THE TOLERANCE BAND

We also conducted relevant experiments on the SMAR loss that controls MRD without employing a tolerance band. Specifically, we investigated two scenarios: one that encourages a significantly enhanced modality distinction in the routing probability distribution, and another that promotes modality-agnostic routing probabilities.

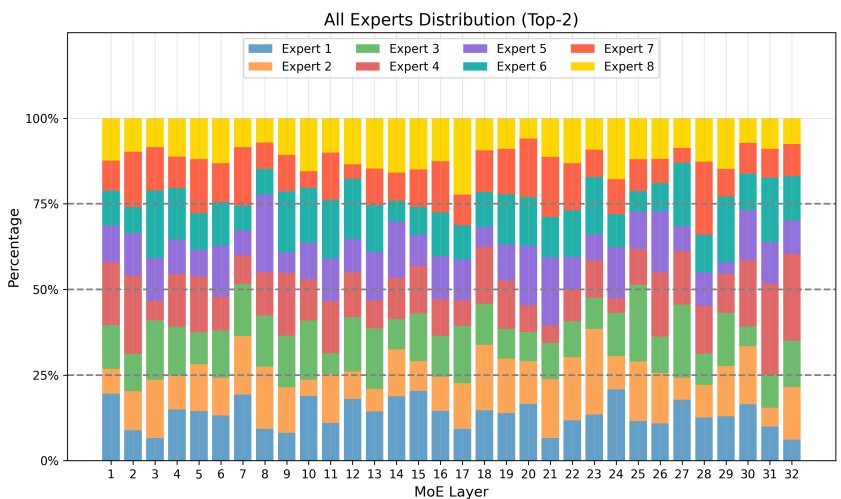

Figure 9: All Experts Usage Proportions for Baseline VITA model with SMAR loss.

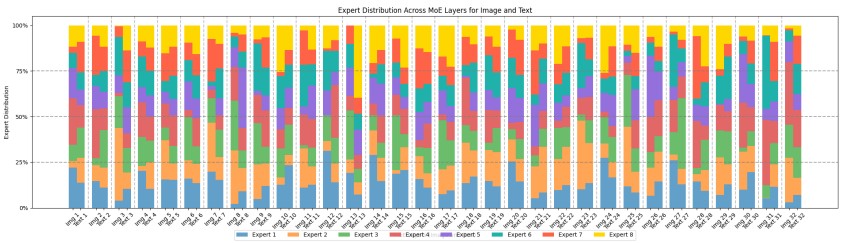

Figure 10: **With the $[d_{min}, d_{max}]$ set to [1.5, 2.0], experts distribution across different modalities on SQA-IMG task**

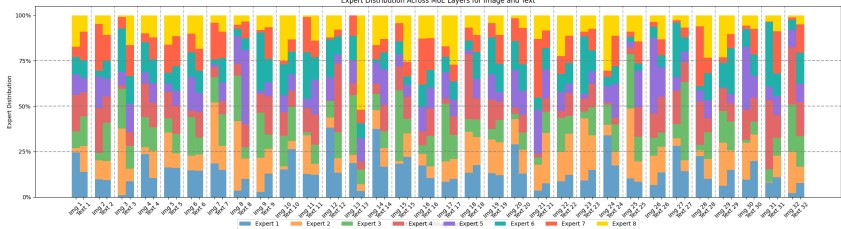

Figure 11: **With the $[d_{min}, d_{max}]$ set to [1.5, 2.0], experts distribution across different modalities on POPE task**

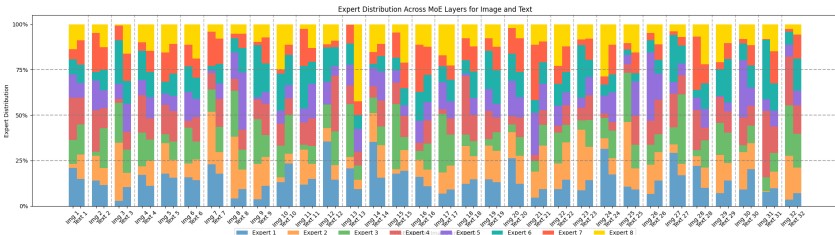

Figure 12: **With the $[d_{min}, d_{max}]$ set to [1.5, 2.0], experts distribution across different modalities on TextVQA task**

In our previous experiments, we observed that setting the MRD target value either too low or too high, without the constraint of load-balancing loss, often leads to routing collapse. Therefore, we incorporated the load-balancing loss and sought to identify the optimal trade-off parameters.

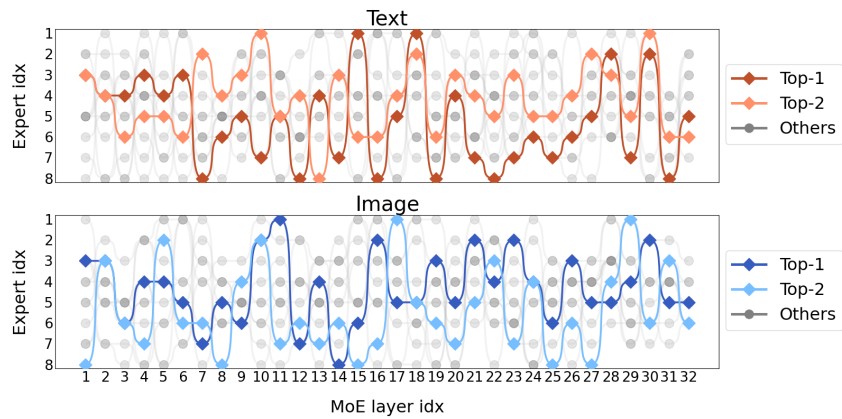

Figure 13: **With the $[d_{min}, d_{max}]$ set to [1.5, 2.0], experts activated pathways on SQA-IMG task**

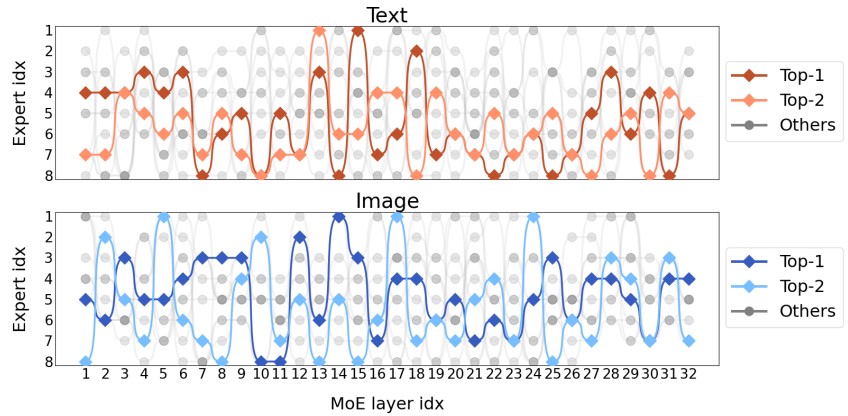

Figure 14: **With the $[d_{min}, d_{max}]$ set to [1.5, 2.0], experts activated pathways on POPE task**

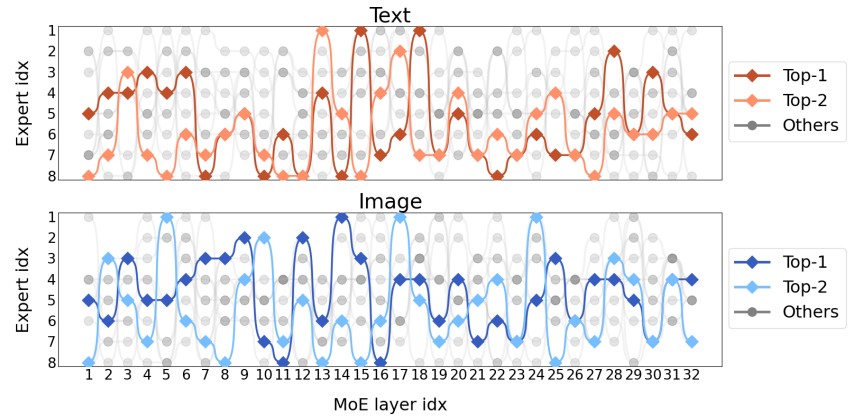

Figure 15: **With the $[d_{min}, d_{max}]$ set to [1.5, 2.0], experts activated pathways on TextVQA task**

The experimental results indicate that encouraging modality separation in the routing probabilities more effectively facilitates the development of models that achieve a favorable balance between multimodal performance and language capabilities, as shown in Table 8.

For the approach that encourages modality fusion, we only present the currently optimal parameter set. This decision is based on the results illustrated in Figure 9 in the main text, as well as our

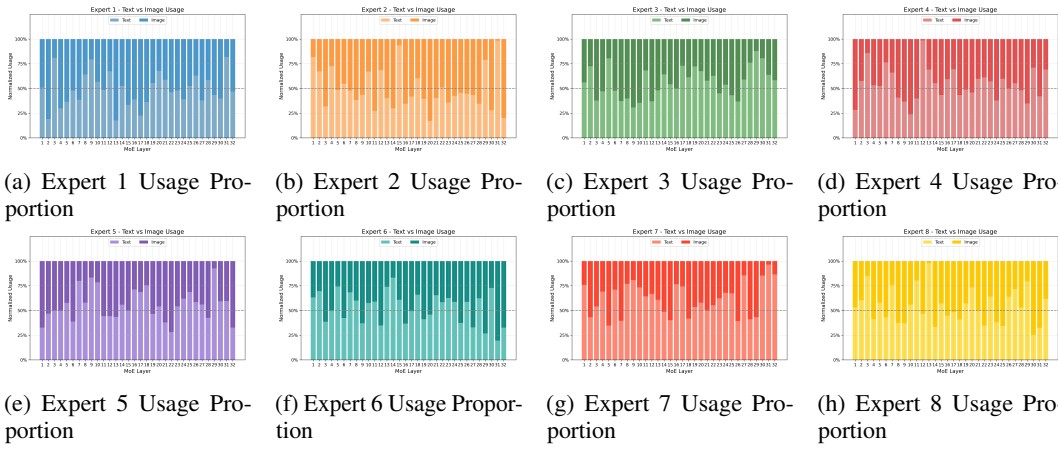

(a) Expert 1 Usage Proportion

(b) Expert 2 Usage Proportion

(c) Expert 3 Usage Proportion

(d) Expert 4 Usage Proportion

(e) Expert 5 Usage Proportion

(f) Expert 6 Usage Proportion

(g) Expert 7 Usage Proportion

(h) Expert 8 Usage Proportion

Figure 16: Experts Usage Proportions for Baseline VITA model

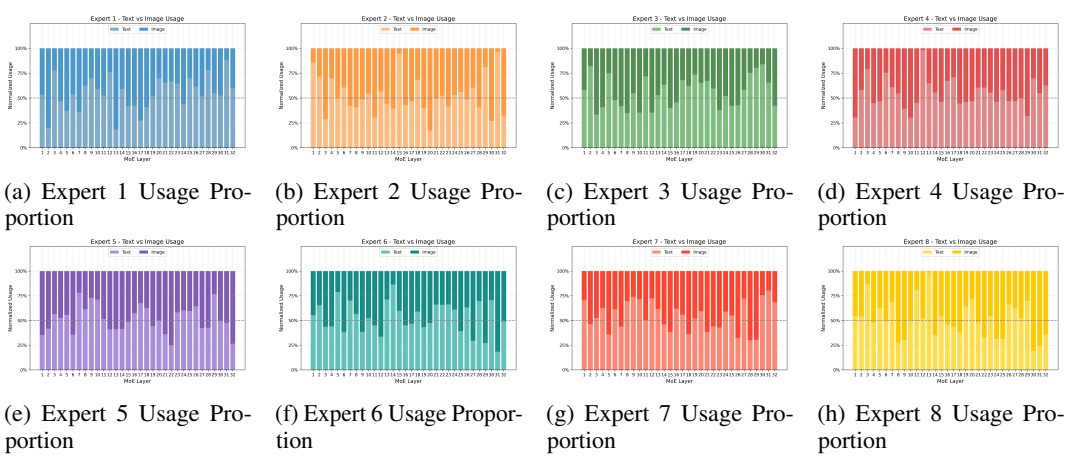

(a) Expert 1 Usage Proportion

(b) Expert 2 Usage Proportion

(c) Expert 3 Usage Proportion

(d) Expert 4 Usage Proportion

(e) Expert 5 Usage Proportion

(f) Expert 6 Usage Proportion

(g) Expert 7 Usage Proportion

(h) Expert 8 Usage Proportion

Figure 17: Experts Usage Proportions for Baseline VITA model with load-balancing loss

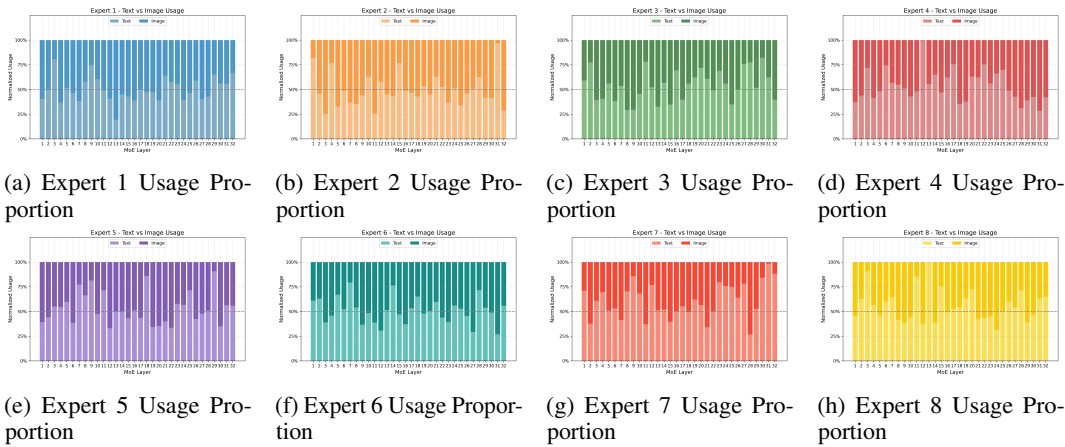

(a) Expert 1 Usage Proportion

(b) Expert 2 Usage Proportion

(c) Expert 3 Usage Proportion

(d) Expert 4 Usage Proportion

(e) Expert 5 Usage Proportion

(f) Expert 6 Usage Proportion

(g) Expert 7 Usage Proportion

(h) Expert 8 Usage Proportion

Figure 18: Experts Usage Proportions for Baseline VITA model with SMAR loss

testing experience with multiple parameter configurations in the modality separation experiments. Specifically, when the $\beta$ is set to 0.0001, a favorable trade-off is achieved without causing routing collapse.

| MRD Target | $\alpha$ | $\beta$ | Multimodal Capabilities | | | | Language Capabilities | | | |
|---|---|---|---|---|---|---|---|---|---|---|
| | | | MME | GQA | SQA-IMG | TextVQA | MMLU | BBH | GSM8K | MBPP |
| $+\infty$ | 0.02 | 0.01 | 1621.38 | 62.39 | 72.78 | 69.02 | 66.56 | 59.16 | 49.28 | 22.60 |
| $+\infty$ | 0.02 | 0.001 | 1599.85 | 62.28 | 72.38 | 69.58 | 68.45 | 62.06 | 56.79 | 20.40 |
| $+\infty$ | 0.02 | 0.0001 | 1630.75 | 62.66 | 73.82 | 69.66 | 67.03 | 61.12 | 56.10 | 27.40 |
| 0 | 0.02 | 0.0001 | 1604.13 | 62.49 | 74.47 | 69.71 | 68.25 | 62.29 | 56.41 | 12.20 |

Table 8: **Comparison under varying MRD targets and trade-off coefficients between the load-balancing loss and SMAR loss on multimodal benchmarks and language benchmarks.** Where $\alpha$ and $\beta$ are hyper-parameters controlling the relative strength of the auxiliary terms: $\mathcal{L}_{\text{total}} = \mathcal{L}_{\text{main}} + \alpha \mathcal{L}_{\text{balance}} + \beta \mathcal{L}_{\text{SMAR}}$ .

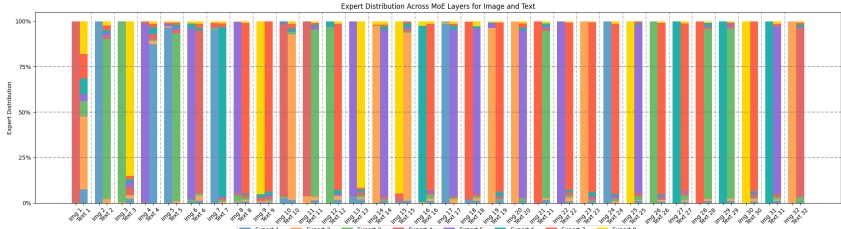

Figure 19: With the MRD Target set to $+\infty$ and $\beta$ set to 0.01, experts distribution across different modalities

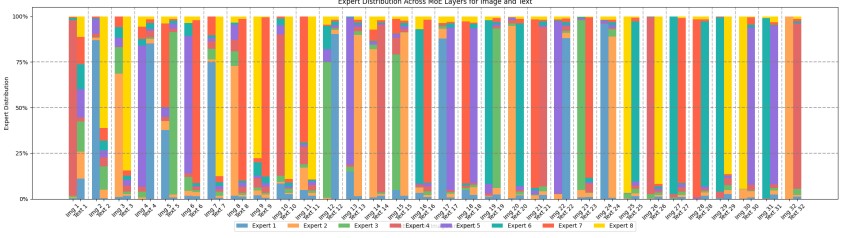

Figure 20: With the MRD Target set to $+\infty$ and $\beta$ set to 0.001, experts distribution across different modalities

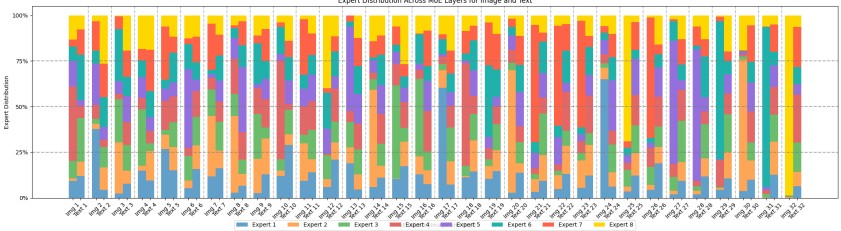

Figure 21: With the MRD Target set to $+\infty$ and $\beta$ set to 0.0001, experts distribution across different modalities

When the $\beta$ exceeds 0.0001, severe routing collapse occurs at every layer as shown in Figure 19, Figure 20. We also observe that when the $\beta$ is set to 0.0001, the approach encouraging increased MRD still induces routing collapse of visual tokens in the final two layers, whereas the approach promoting decreased MRD does not as shown in Figure 21, Figure 22. This phenomenon may be attributed to the fact that encouraging an increase in MRD imposes a constraint towards positive infinity, exerting a stronger regularization effect, whereas encouraging a decrease in MRD sets the target constraint to zero, resulting in a comparatively weaker enforcement. Consequently, the optimal trade-off parameters for these two approaches are not entirely aligned.

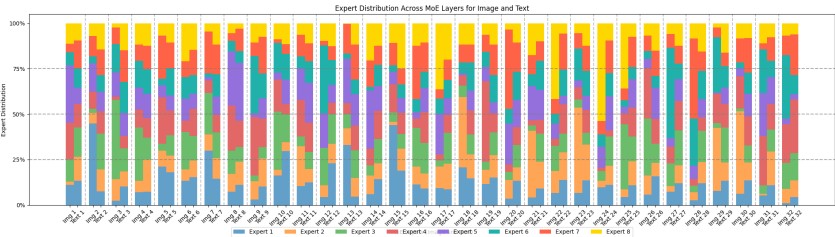

Figure 22: With the MRD Target set to 0 and $\beta$ set to 0.0001, experts distribution across different modalities

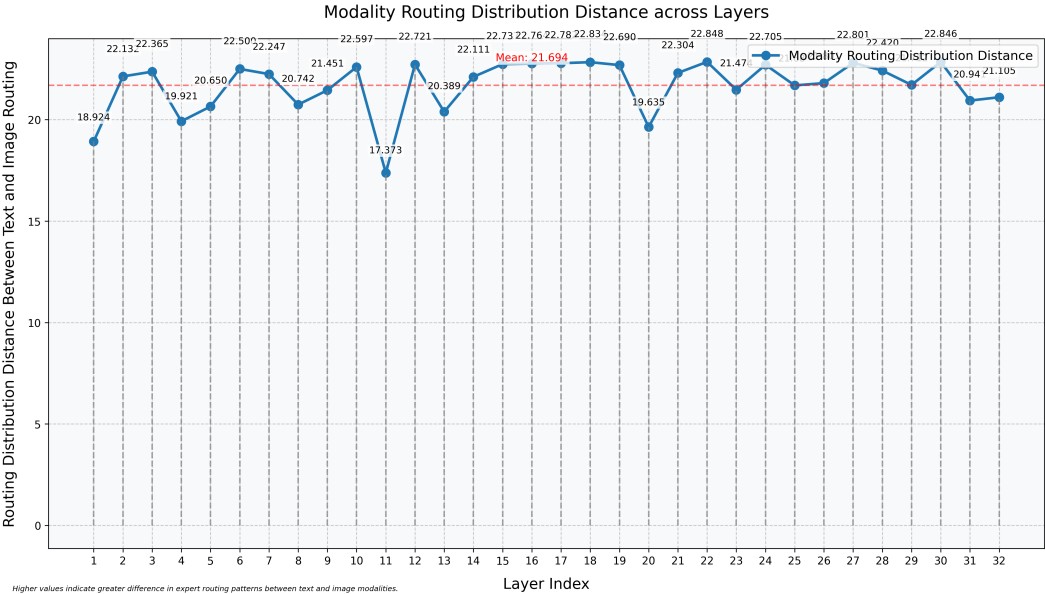

Figure 23: With the MRD Target set to $+\infty$ and $\beta$ set to 0.01, the MRD curve across every layer.

The experimental results indicate that both the MRD control method utilizing a tolerance band and the approach balancing the SMAR loss with the load-balancing loss support the conclusion that encouraging an increase in MRD more readily leads to optimal performance across multiple modalities. **This insight appears to provide valuable guidance for the design of other MoE-MLLMs: maintaining an appropriate modality routing separation strategy is beneficial.** We validated this hypothesis through continuous experimentation by naturally controlling the variations in MRD.

## D  THE USE OF LARGE LANGUAGE MODELS

Regarding the use of large language models, we used them solely for linguistic polishing of the writing. They made no other substantive contributions to the work.

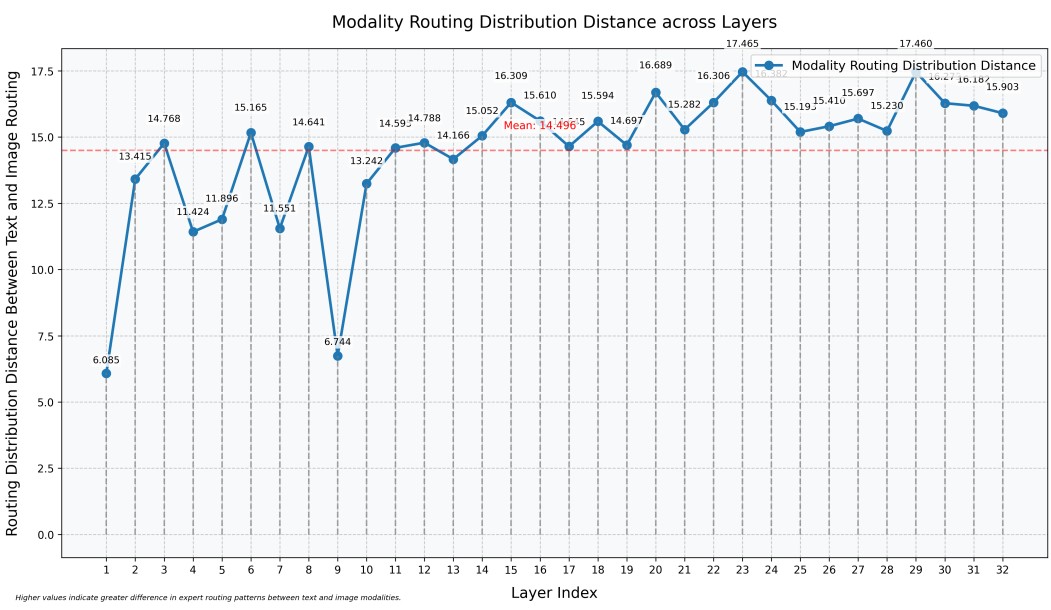

Figure 24: With the MRD Target set to $+\infty$ and $\beta$ set to 0.001, the MRD curve across every layer.

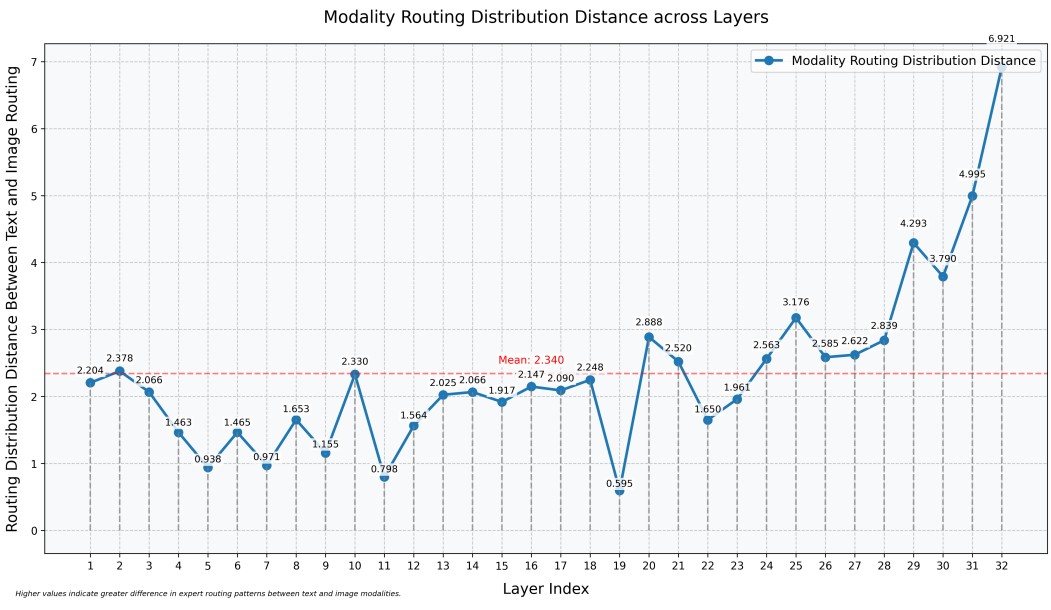

Figure 25: With the MRD Target set to $+\infty$ and $\beta$ set to 0.0001, the MRD curve across every layer.

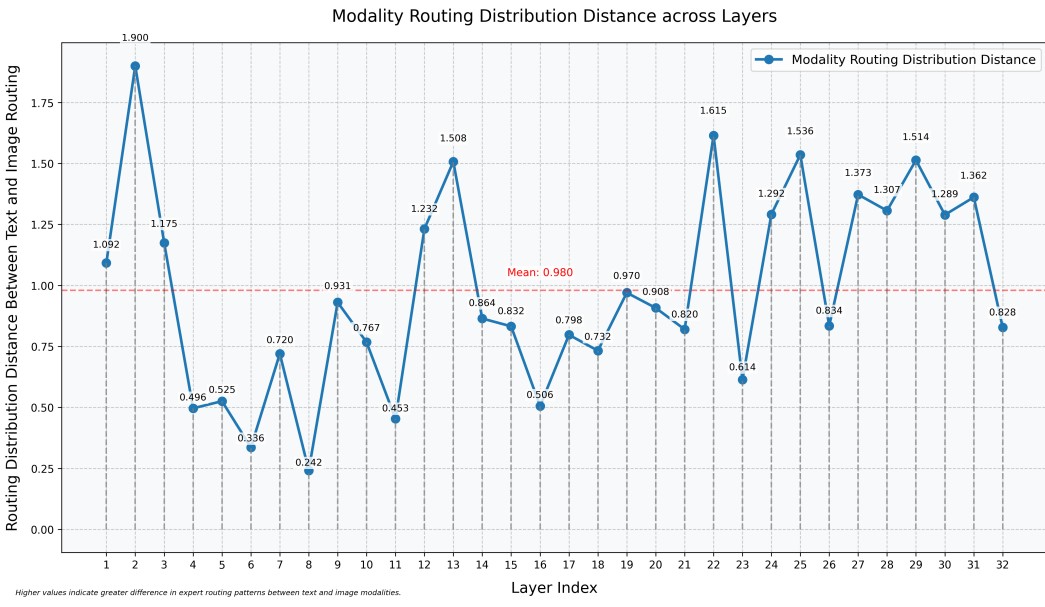

Figure 26: With the MRD Target set to 0 and $\beta$ set to 0.0001, the MRD curve across every layer.

