# OpenReview forum: "SMAR: Soft Modality-Aware Routing Strategy for MoE-based Multimodal Large Language Models Preserving Language Capabilities"
_ICLR.cc/2026/Conference — ICLR 2026 Conference Withdrawn Submission_

### Official Review · Reviewer_EVSU · 2025-10-25

**Soundness:** 2
**Presentation:** 3
**Contribution:** 2
**Rating:** 4
**Confidence:** 4

**Summary:**

This paper proposes Soft Modality-Aware Routing (SMAR), a regularization technique for Mixture-of-Experts (MoE) based multimodal large language models (MLLMs) to preserve language capabilities during multimodal training. The key contributions include: (1) introducing Modality Routing Distribution (MRD) as a novel metric to characterize routing patterns across different modalities, (2) designing a tolerance band-based loss function using Kullback-Leibler divergence to control expert specialization, and (3) demonstrating language capability retention of 86.6% with only 2.5% pure text data, compared to 81.6% for the baseline without auxiliary loss.

**Strengths:**

The paper introduces a novel perspective on analyzing MoE routing behavior through MRD, which provides valuable insights into modality-specific expert specialization. The tolerance band approach for controlling expert differentiation is creative and theoretically motivated. The core methodology is clearly explained with appropriate mathematical notation. The MRD concept is well-motivated and the SMAR loss formulation is understandable. Addressing language capability degradation in multimodal training is an important problem. The proposed approach offers a potential solution without requiring architectural modifications or extensive pure text data.

**Weaknesses:**

**Misleading Performance Claims**: The paper's main performance claims are problematic. The authors compare their MoE-based method against dense models (LLaVA-1.5) to claim improvements of 5.4%, 7.0%, 7.4%, etc., which is fundamentally unfair due to architectural differences. When compared against fair baselines with the same architecture, the multimodal improvements are marginal (typically <2%) and sometimes negative (e.g., VQAv2: 82.5→82.4).

**Questionable Baseline Quality**: The baseline model shows anomalously poor performance on MBPP (10.4 vs. 49.0 for base Mixtral), suggesting potential training issues. This 70%+ degradation is inconsistent with other language metrics (85-94% retention) and raises questions about experimental validity. The dramatic "improvement" from SMAR (10.4→28.4) appears more like fixing a broken baseline than algorithmic innovation.

**Incomplete Experimental Reporting**: Table 5 (ablation study) only reports language capabilities, omitting multimodal metrics entirely

**Statistical Issues**:
  - The paper contains a basic reporting error: it claims results on six benchmarks but lists only five improvement percentages. For example, Section 4.3 ("RESULTS") states: "Specifically, on SQAI, MME, MMBench, MM‑Vet, VQAT, and VQAv2, our model achieves performance gains of 5.4%, 7.0%, 7.4%, 12.8%, and 3.0%, respectively, over LLaVA‑1.5‑13B," which is inconsistent.
  - No significance testing or confidence intervals provided
  - Results may not be statistically meaningful given small effect sizes AND inconsistent patterns across benchmarks. Table 1 shows that SMAR's effects are highly inconsistent across different benchmarks, with both improvements and degradations:

Baseline vs Baseline w/ SMAR:

  VQAv2:    82.5 → 82.4  (-0.1, decline)

  GQA:      62.2 → 62.4  (+0.2, minimal gain)

  VizWiz:   53.7 → 55.1  (+1.4, modest gain)

  SQAI:     74.6 → 75.5  (+0.9, modest gain)

  VQAT:     69.6 → 69.2  (-0.4, decline)

  POPE:     86.8 → 86.6  (-0.2, decline)

  MME:      1634.7 → 1638.8 (+4.1, minimal gain)

This mixed pattern of gains and losses suggests that:
  1. The method lacks consistent effectiveness across different task types
  2. Improvements may be within noise levels rather than systematic algorithmic gains
  3. The overall "improvement" narrative is misleading when individual benchmarks show contradictory trends
  4. Cherry-picking favorable comparisons (e.g., emphasizing gains vs LLaVA-1.5 while downplaying mixed results vs fair baselines) may be masking the true performance profile

The lack of a coherent improvement pattern across related benchmarks (e.g., VQAv2 declining while other VQA tasks improve slightly) raises questions about whether SMAR provides genuine algorithmic benefits or merely redistributes performance across different evaluation scenarios.

**Trade-off Analysis Missing**: The method exhibits clear language-multimodal performance trade-offs (evident in Table 3), but the paper fails to acknowledge or analyze this limitation. This omission is misleading for practitioners who need to understand the full implications of the approach.

**Questions:**

1. **Baseline Validity**: Can you explain why the baseline MBPP performance (10.4) is so dramatically lower than the base Mixtral performance (49.0) in Table 2, while other language metrics show much smaller degradations? What specific training issues might have caused this anomaly?
2. **Fair Comparison**: Why compare against dense models (LLaVA-1.5) rather than focusing on improvements over fair baselines with identical architectures? Could you provide a more balanced discussion of the actual gains when architectural advantages are controlled?
3. **Complete Ablation Results**: Table 5 is missing multimodal performance metrics. Can you provide the complete ablation results showing how modality-aware bias and load-balancing loss affect both language and multimodal capabilities?
4. **Trade-off Quantification**: The Table 3 results suggest language improvements come at multimodal performance costs. Can you quantify this trade-off and provide guidance on optimal operating points for different application scenarios?
5. **Threshold Selection**: The optimal thresholds [1.5, 2.0] were chosen based on "best overall language score," but the selection criteria are unclear. How was "overall" defined, and why weren't other metrics considered in the selection?

While the core idea has merit, the experimental validation contains significant flaws that undermine the paper's claims. The work would benefit from more rigorous experimental design, complete reporting of results including trade-offs, and honest discussion of limitations.

---

### Official Review · Reviewer_JyHr · 2025-10-28

**Soundness:** 3
**Presentation:** 2
**Contribution:** 2
**Rating:** 4
**Confidence:** 4

**Summary:**

This paper introduces Soft Modality-Aware Routing (SMAR), a KL-divergence based loss that shapes modality routing distributions of MoE experts to explicitly control modality specialization without architecture changes, trying to preserve language ability while maintaining strong multimodal performance.

**Strengths:**

1.	Clear and practically meaningful motivation: The paper pinpoints a real, widespread issue in LVLMs—when integrating visual capabilities via multimodal data, the underlying language competence of the LLM is degraded; addressing this is crucial for real-world deployment of LVLMs.
2.	Diagnostic “metric” for modality-wise routing: The authors propose a novel MRD “metric” to evaluate routing probability distributions across modalities, providing a useful lens for analyzing routing strategies in MoE-based multimodal models.
3.	Soft, controllable modality-aware routing: The authors introduce trainable per-modality biases and a tolerance-banded symmetric-KL constraint between vision/text routing distributions, enabling controlled expert specialization without hard partitions while maintaining load balance and task performance.
4.	In certain specific areas—such as code generation—the proposed improvements yield substantial gains in model performance.
5.	The authors conduct extensive visualizations of expert activations and routing characteristics, which are valuable for analyzing internal signal flow in sparse multimodal large models.

**Weaknesses:**

1.	Insufficient experimentation: Although the authors claim improvements on multiple metrics, the experimental section does not provide sufficient evidence. For example, in Table 1 across nine multimodal datasets, the proposed method is best on only four; on text datasets, it is best on only 4/8. In Table 2, across six datasets, only two are best. While the mean reportedly improves from 81.6% to 86.6%, if MBPP is excluded, the mean gain drops from 5% to 0.4%. If the improvements hold primarily on domain-specific datasets, the evaluation and claims should focus on that class of datasets.
2.	Missing ablations: For the tolerance band (Table 4), the ablation varies only the band’s location, not its width, and does not explore values to the right of [1.5, 2.0]. For the modality-specific bias ablation (Table 5), an experiment without the modality-specific bias but retaining the Load-Balancing Loss is missing; thus, the gains in the final row cannot be ruled out as stemming from the Load-Balancing Loss.
3.	Imprecise terminology: The mentioned “MRD distance” defined via symmetric KL divergence does not satisfy the axioms of a mathematical metric (triangle inequality). The wording should be revised.

**Questions:**

1.	In Appendix C, authors state that encouraging MRD toward a target value has clear advantages over using constraints of +∞ or 0. However, the benefits of a tolerance band are not discussed. What advantages does a tolerance band offer compared with supervising toward a single exact value?
2.	Prior work suggests that the roles of visual and language tokens differ across layers in LVLMs. Is it feasible to use different hyperparameters d across layers to constrain the MRD distance—or even make d adaptive?
3.	In Table 5, why is [1.0, 1.5] used as the range for d? This differs from the other experiments.

---

### Official Review · Reviewer_FkpW · 2025-10-31

**Soundness:** 2
**Presentation:** 3
**Contribution:** 2
**Rating:** 4
**Confidence:** 3

**Summary:**

This work introduces the Soft Modality-Aware Routing Strategy, which controls the routing of experts across modalities. The author showed that the language capabilities are better preserved while maintaining good multimodal performance. Overall, the writing and organization of the paper are good.

**Strengths:**

1) The paper studies and proposes a metric to quantify the routing strategies in MoE-based multimodal models.
2) The paper proposes SMAR to control expert modality differentiation.
3) The experiment and ablation are well designed with detailed analysis to study SMAR.

**Weaknesses:**

1) The SMAR does not generally improve multimodal capabilities, although it retains language capabilities better. Why?
2) There is a lack of comparative analysis with works mentioned in the intro and related works sections to show how this work challenges SOTA.
3) The abstract mentioned that 2.5% pure text was used, while the conclusion mentioned "without additional pure text"?

**Questions:**

1) Line 416, why only layer 13 almost all text token, but not the other layers?

---

### Official Review · Reviewer_UCfG · 2025-11-02

**Soundness:** 2
**Presentation:** 3
**Contribution:** 2
**Rating:** 4
**Confidence:** 4

**Summary:**

This paper proposes Soft Modality-Aware Routing (SMAR), a novel regularization technique for Mixture-of-Experts (MoE)-based multimodal large language models (MLLMs). The method addresses the challenge of maintaining strong language capabilities while adapting pretrained MoE models to multimodal tasks. The SMAR method uses Kullback-Leibler (KL) divergence to control routing probabilities across different modalities, encouraging expert specialization without heavily relying on textual data. The approach demonstrates strong multimodal performance and achieves 86.6% retention of language capabilities with only 2.5% pure text, outperforming baselines. The paper provides experiments and ablation studies to validate the proposed method on several multimodal benchmarks.

**Strengths:**

- The proposal introduces a unique method, SMAR, to manage expert specialization in MoE-based multimodal models. It creatively uses KL divergence to control routing across modalities, enhancing language capability retention without requiring architectural changes. The idea of soft modality-aware routing is innovative and addresses the crucial challenge of balancing modality differentiation with language performance.
- The experimental setup is comprehensive, and the paper uses relevant and established benchmarks to evaluate the proposed method. The ablation studies and comparison with baselines provide a solid understanding of the method’s performance in both multimodal and language tasks. The authors also present clear details of the model architecture and training strategy, ensuring reproducibility.
- The paper is well-structured and clearly written. The method is explained in detail, and the experimental results are presented in a clear manner with sufficient visual aids (e.g., tables and figures). The paper avoids unnecessary jargon and communicates complex ideas in an accessible way.

**Weaknesses:**

- The paper uses VITA and MoE-LLaVA as the base models in the experiments. They are now considered outdated. The validity of the proposed method on these older models might not be representative of its potential on more advanced architectures.
- The paper compares SMAR with load-balancing loss, but it lacks sufficient details about how load-balancing loss is applied in the experiments. There are no descriptions of the training parameters, settings, or any explanation of why this method is being used as a baseline.
- The benchmarks used in this paper are outdated, both in terms of pure-text and multimodal evaluations. The current benchmarks primarily focus on simple question answering (QA) tasks and lack a comprehensive assessment of reasoning abilities. Reasoning is crucial for evaluating the true strength of a model’s language capabilities, especially when tackling more complex tasks, which is essential for validating the core claim of this paper. Similarly, newer multimodal benchmarks incorporate more challenging tasks, such as multi-image inputs and video understanding, which better assess the model’s visual capabilities. Updated multimodal benchmarks, with their increased visual challenges, would provide a more thorough examination of how the proposed method impacts the model’s visual reasoning abilities.
- The proposed method focuses on a specific configuration of the MoE model (e.g., Mixtral). However, it does not discuss how well SMAR would scale to even larger models or more complex multimodal tasks. Given the rapid development of MoE-based architectures, it is crucial to understand how this method might behave in larger-scale models.

**Questions:**

- Could you elaborate on the specific training parameters and settings used with load-balancing loss?
- How would SMAR perform on more recent pure-text reasoning benchmarks that focus on more complex reasoning tasks?
- How would SMAR scale when applied to larger and updated models or more complex multimodal tasks?

---

### Note · Authors · 2026-01-06

I have read and agree with the venue's withdrawal policy on behalf of myself and my co-authors.